# ROBUST COMPONENT DETECTION FOR FLEXIBLE MANUFACTURING:
# A DEEP LEARNING APPROACH TO TRAY-FREE OBJECT RECOGNITION UNDER VARIABLE LIGHTING

## ABSTRACT

Flexible manufacturing systems in Industry 4.0 require robots that can handle objects in unstructured environments without rigid positioning constraints. This paper presents a computer vision system that enables industrial robots to detect and pick up pen components in arbitrary orientations without the need for structured trays, while maintaining robust performance under varying lighting conditions. We implement and evaluate a Mask R-CNN-based approach in a complete pen production line, addressing three key challenges: object recognition without positional constraints, robustness to extreme lighting changes, and reliable performance with cost-effective cameras. Our system achieves 95% recognition accuracy under diverse lighting conditions and eliminates the need for structured component placement, resulting in significant improvements in manufacturing flexibility and overall robustness. This approach has been validated through extensive experiments under four distinct lighting scenarios. These results demonstrate its practical applicability for real-world industrial deployment.

## 1 INTRODUCTION

In recent years, industry intelligence, in the form of the Fourth Industrial Revolution (Industry 4.0), has introduced advanced manufacturing techniques by integrating the Internet of Things and artificial intelligence into manufacturing environments (Ghobakhloo, 2020). Industry 4.0 has been considered a competitive approach in the industry over the last decade, especially in European Union member states, and has become one of the key factors in improving productivity and competitiveness (Lu, 2017). The term "4.0" has been used for various areas such as Services 4.0 (Lu, 2017), Agriculture 4.0 (Buer et al., 2018),Agribusiness 4.0 (de Macedo et al., 2018), Healthcare 4.0 (Chanchaichujit et al., 2019), and Logistics Winkelhaus et al.,2020. In this context, the need for intelligent perceptual systems is increasingly felt, and machine vision, as one of the main pillars of Industry 4.0, plays the role of the "digital eye" of production lines, which is very important for visual inspection, part identification, and automation (Golnabi & Asadpour, 2007). The use of machine vision methods has paved the way for intelligent production in robotic processes. Thus, with the increasing importance of competition in the use of Industry 4.0, visual computing techniques are considered more than ever (Wuest et al., 2016; Alayed et al., 2024).

Machine vision enables automated inspection, defect detection, and robot guidance, and when combined with artificial intelligence, significantly increases the flexibility and quality of manufacturing processes. (Profili et al., 2024). Machine vision systems consist of various components, including vision sensors, lighting, and computers, which together enable these systems to provide a visual representation of their environment. Despite advances in machine vision technology, such as improved sensors, lighting, and processing power, further progress is still needed to overcome existing challenges (Bagheri et al., 2020). In the context of Industry 4.0, one of the fundamental challenges in implementing smart manufacturing systems is the ability to accurately recognize objects and parts in complex environmental conditions. These conditions include changing lighting, random positions of parts, and the presence of noise in the image (Trampert et al., 2025). One real-world example that

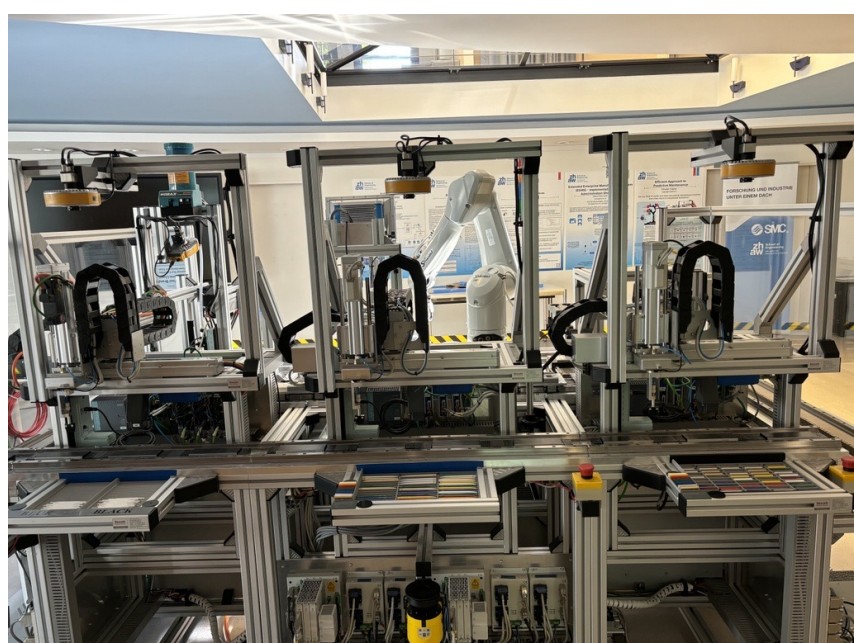

Figure 1: Complete pen production line showing integrated robotic stations

illustrates these challenges well is the industrial pen production line. There are three major problems in this process: first, the need to arrange parts in neat trays, which increases preparation time and reduces flexibility; second, the high sensitivity of cameras to ambient light changes, which reduces detection stability; and third, light reflections, which lead to false detections. This example highlights the importance of developing solutions that are resistant to light changes and irregular arrangements of parts in Industry 4.0.

To overcome these problems, object detection techniques have been developed in the field of machine vision, which are divided into two main categories: traditional methods based on machine learning and modern methods based on deep learning. In traditional methods, feature engineering plays a key role; For example, the generalized Hough transform was designed to extract geometric features (Ballard, 1981), the Harris corner detector was designed to detect corner landmarks (Harris & Stephens, 1988), and scale- and rotation-invariant methods were designed to cope with dimensional and angular variations (Lowe, 2004). Although these methods are useful in many applications, they have limitations when dealing with complex backgrounds, strong light changes, and highly diverse objects. In contrast, deep learning methods learn optimal features directly from the data without the need for manual feature engineering and achieve higher accuracy in object recognition. Architectures such as R-CNN (Kühlechner et al., 2025); Fast R-CNN (Leng et al., 2025), Faster R-CNN (Wang et al., 2025), YOLO (Schneidereit et al., 2025), SSD (Kang et al., 2025), and Mask R-CNN (He et al., 2017). in real Industry 4.0 environments are able to identify manufactured parts in disordered states and under varying lighting conditions with high accuracy, thus paving the way for fully automated and flexible manufacturing.

The pen production line is a challenging example of machine vision applications in Industry 4.0 due to the variety of parts, sensitivity to correct alignment, and high speed expected. In addition, the presence of shiny surfaces, such as the metal or plastic body of the pen, causes light reflections that can lead to false detections and doubles the need for algorithms that are robust against noise and lighting changes. Considering the challenges of Industry 4.0 and machine vision-based methods, this study focuses on the fountain pen production line and presents a Mask R-CNN system for tray-free part recognition under variable lighting conditions, as illustrated in Figure 1.

In this study, the Tray-Free Object Recognition Under Variable Lighting is performed which demonstrates the success of applying Mask R-CNN for robust part recognition in flexible manufacturing environments and addresses three key challenges in Industry 4.0 manufacturing:

- Eliminating structured positioning requirements,

- Resisting lighting changes,
- Reliable performance with cost-effective camera systems.

Key findings of this research include:

- Our approach eliminates the need for structured component trays, resulting in a significant improvement in manufacturing flexibility.
- The system maintains an average recognition accuracy of 93.4% under varying lighting conditions, addressing one of the main limitations of traditional vision systems.
- Successful deployment on a full pen production line validates the practical application of deep learning approaches in real-world industrial environments. The system's ability to handle arbitrary component orientations while maintaining high accuracy is a significant advancement in flexible manufacturing capabilities.
- The successful integration of AI-based vision systems with existing industrial infrastructure provides a roadmap for similar implementations in various manufacturing domains.

The paper is organized as follows. Section 2 describes the materials and methods, including the Industry 4.0 demonstration system and the Mask R-CNN-based deep learning approach. Section 3 presents the experimental results, including the training performance, evaluation criteria, and analysis under different lighting conditions. Section 4 compares the proposed method with state-of-the-art approaches and provides detailed performance analysis. Finally, Section 5 concludes the paper and discusses potential directions for future research..

## 2 Materials and Methods

### 2.1 Industry 4.0 Demonstrator System

The Industry 4.0 demonstrator is an industrial robot for demonstrations to companies with different purposes and proves the capabilities and practical implementations made. In comparison to many dangerous arm robots that cannot recognize humans around them and may collide with humans, the Industry 4.0 demonstrator is a safe robot that humans can interact with in a safe environment. The robot has special sensors that can perform its operations if they are not green, but when a person approaches it, the lights turn red and the robot stops working. In this research, experiments were designed and implemented in an Industry 4.0 demonstrator to improve the performance of a fully automated production line for pen assembly. The production line consists of several robots and workstations that operate in coordination, enabling flexible and customized production (Figure 1). The production process is as follows:

**Customer Interface:** The customer enters the display program by scanning a QR code and registers their order. He can choose the body color, cap color, and the desired text to be printed on the pen.

**Order Processing:** The order is stored in the database and then sent to the main computer of the production line via a script to begin the assembly process.

**Delta Robot Station:** An empty carrier enters the production line. This carrier moves between different stations during the process so that the pen parts are placed on it in order. The carrier is first transferred to the Delta robot station. The Delta robot is responsible for the *Pick-and-Place* task of placing the springs and primary mechanical parts on the carrier. At this stage, the order number is recorded on the carrier so that it can be synchronized with the customer's order during the assembly process.As shown in Figure 2(a), the Delta robot station performs the operation of picking and placing springs and thrust parts and acts as a key point in the automation of the production line.

**Refill Station:** Then the carrier is transferred to the "Refill" station. As it is shown in Figure 2(b), The robot at this station reads the order information and based on the selected color (blue or black), picks up the appropriate core and places it on the carrier.

**Barrel Station:** At the next station, the carrier enters the "body" station. The robot at this station uses machine vision to identify the correct body color from the tray and place it on the carrier. Barrel station is demonstrated in Figure 2(c).

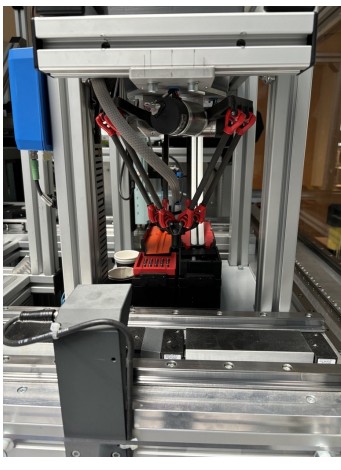 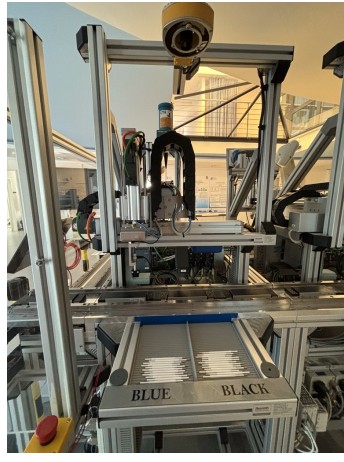 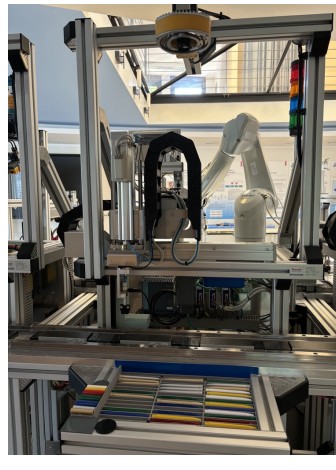

(a)           (b)           (c)

Figure 2: Key stations in the production line: (a) Delta station; (b) Refill station; (c) Body station.

**Cap Station:** At the next stage, the carrier is transferred to the "cap" station. Portal robot number 2 reads the cap color from the order, selects the appropriate cap from the tray, and places it on the carrier. At this stage, the caps must be positioned correctly so that the robot gripper can pick them up.

**Quality Inspection:** After the cap is installed, an inspection stage is performed to ensure that all parts are present in the carrier. If a part is defective, the carrier is sent back to the relevant station to be repaired.

**Assembly Station:** When the carrier is complete, it enters the final assembly station. The main robot (STAUBLI) assembles the parts and guides the carrier to the printer to print the text. If there are multiple orders at the same time, the system decides which printer slot to use and sends the order number to the printer.

**Printing Station:** The printer prints the customer's selected text on the pen body and sends a completion message to the system.

**Final Assembly:** Finally, the assembly robot picks up the complete pen from the carrier and delivers it to the customer.

As discussed in Section 1, the key problem is to remove the tray and enable robust orientation recognition. Traditional image processing algorithms have been tested but have yielded disappointing results. Given the diverse orientations in different situations, a machine learning approach is necessary to solve this complex problem. Therefore, we used a deep neural network, Mask R-CNN, to recognize the mask and orientation of each pen.

## 2.2 PROPOSED METHOD AND EXPERIMENTAL SETUP

**Network Architecture:** Mask R-CNN is a convolutional neural network that generates boundaries and segmented regions for each object in an image. This deep neural network is built based on Feature Pyramid Network (FPN) and ResNet101. Mask R-CNN is implemented on Python, Keras, and Tensorflow. We used Mask R-CNN along with the open-source library Facebook Detectron2. Furthermore, due to the need for a GPU to implement the deep neural network, we implemented the project on Google Colab, which provides a GPU. The programming language for the project is Python 3.7, and we use PyTorch. Furthermore, we use Ubuntu in the Colab environment.

**Data Collection:** Images were captured using the IDS UI-3280CP Rev. 2 integrated into the demonstration robot system. The dataset consisted of 87 images of the pens in different orientations and under different lighting conditions, which was used continuously for training and testing. This ensured that the training and deployment environments were consistent and minimized potential performance gaps.

Table 1: Summary of dataset composition including number of images and objects under each lighting condition.

| Dataset Component | Images | Total Objects |
|---|---|---|
| Training Set | 8 | 87 |
| Mixed barrels/caps | 6 | 71 |
| Individual components | 2 | 16 |
| | | |
| Test Set | 102 | $> 1,020$ |
| Intensive lighting | 20 | $> 200$ |
| Dark environment | 20 | $> 200$ |
| Front-lit | 31 | $> 310$ |
| Back-lit | 31 | $> 310$ |

**Data labeling:** The training data was prepared using the LabelMe annotation tool. We manually drew precise boundaries around each pen component in eight sample images, each containing multiple components in different orientations and lighting conditions.

**Training configuration:** The deep convolutional neural network, Mask R-CNN, was trained using Google Colab GPU resources with Python 3.7 and PyTorch in an Ubuntu environment. The training dataset was carefully balanced to include different lighting and layout scenarios and different numbers of components to avoid overgeneralization.

**Testing:** After training Mask R-CNN with labeled data, we tested the model using images captured in similar conditions. The goal was to ensure that the network could accurately recognize the mask and the orientation of the pens without binning, even in different situations. The evaluation was performed using 102 test images in four distinct lighting conditions: (1) high ambient light, (2) dark environment, (3) front-lit conditions, and (4) back-lit scenarios. Each batch consisted of approximately 20 images with an average of more than 10 components per image.

We classified the data into training and testing datasets. We took pictures of pens in different lighting conditions, and in some images, we included several pens, and in others we included fewer pens. Initially, we labeled the boundary of 87 pens, a group of 18, 14, 12, and 2 barrels. A mixed group of 6 caps and 21 barrels. One image with 1 cap alone and another with 1 barrel alone. We chose these 8 images as training data, and the test data can be dynamic with any input we want to find the boundary of the pen in the image. Table 1 summarizes the details of the training and test data.

After training the network on the dataset, we can feed the network any test image, and it will accurately detect each barrel or cap in the image. The previous methods have some challenges. It is difficult to recognize robots in different lighting conditions. Orientation recognition was not possible without any arrangement and tray; Moreover, when the light reflection on the barrels is as a line of light in the middle of the barrel, the previous vision system would incorrectly recognize the barrel as two barrels instead of one barrel with a line of light in the middle. We implement a Mask R-CNN network to address these challenges.

## 3 RESULTS

### 3.1 TRAINING PERFORMANCE

In this section, we report on the research results along with the details of the experiments. We generated a set of labeled training data in different lighting conditions and positions with 22 images and multiple pens within each image. There are multiple pens in different positions in each image, and the advantage of this algorithm is that it can work with fewer datasets than the current datasets with outstanding results. Using the dataset described in Section 2, the model achieved the training performance summarized below. Figure 3, shows that the training accuracy (Chandrasekhar & Peddakrishna, 2023) improves over time and reaches approximately 0.95 in the final stage of training. This shows that the model performs well in detecting the training data. However, for a more robust evaluation, we need to consider additional metrics and statistical analysis.

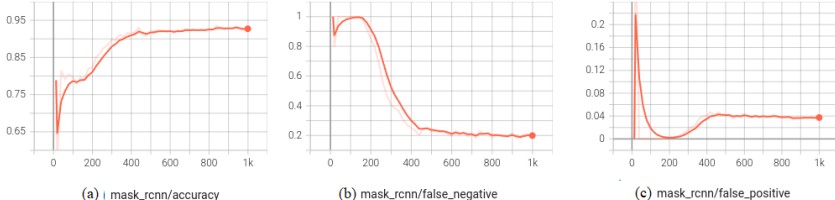

Figure 3: Training performance metrics: (a) accuracy, (b) false negative rate, (c) false positive rate over training.

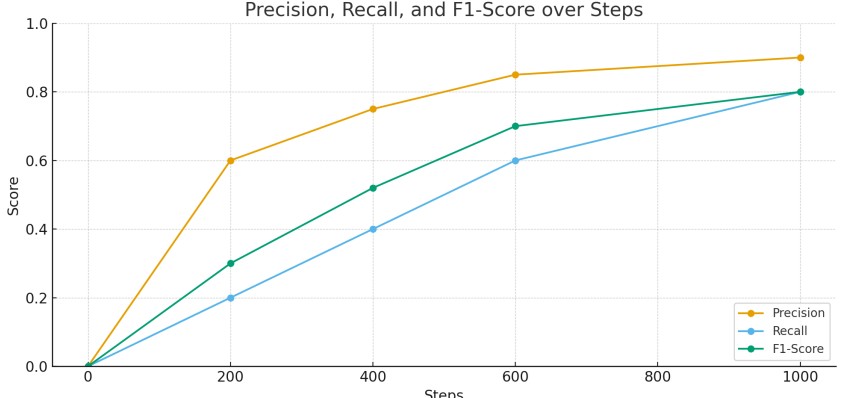

Figure 4: Precision, recall and, F1-score progression during training.

During training, the false negative rate (Alayed et al., 2024) decreases to 0.2. False negatives refer to cases where the model fails to detect pen barrels in the images. A false negative rate of 0.2 indicates that 20% of the real pen barrels are not detected by the model. Furthermore, the false positive rate (Cao et al., 2024), which measures the cases where the model incorrectly identifies an object as a pen barrel when it is not, decreases to 0.04 by the end of training. This low false positive rate indicates that only 4% of non-pen objects are incorrectly classified as pen barrels.

## 3.2 COMPREHENSIVE EVALUATION METRICS

In addition to accuracy, we also analyzed the reference precision, recall, and F1 scores under different lighting conditions (Simeth et al., 2024). The precision and recall and, F1-score metrics in Figure 4. were tracked during training, and the F1 scores were plotted to observe the trade-off between precision and recall, providing a comprehensive assessment of the model's performance. The precision metric, which measures the proportion of correct positive identifications among all positive predictions, showed consistent improvement over training. The recall metric which represents the model's ability to recognize all positives, showed consistent performance across different lighting conditions. The F1 score represents the harmonic mean of precision and recall and provides a single metric and provided a balanced assessment of the model's performance. Our results demonstrate robust performance across a variety of testing scenarios, and the model maintained its effectiveness under varying lighting conditions that previously challenged traditional image processing approaches.

## 3.3 LIGHTING CONDITION ANALYSIS

The model was evaluated under four distinct lighting conditions (as described in Section 2.2). In each condition, the model showed strong performance, achieving an overall accuracy of over 90% despite challenging lighting conditions, confirming its reliability in various real-world scenarios. Testing in four distinct lighting scenarios demonstrated the robustness of the model to environmental variations:

As can be seen in Table 2, in high-intensity ambient lighting, the model maintained a recognition accuracy of 94.2%, successfully handling challenging reflections and shadows that previously caused traditional methods to fail. Low light conditions: In dark environments, the model achieved an accuracy of 92.8% by using learned features that are less dependent on absolute brightness values. Di-

Table 2: Performance comparison under different lighting conditions.

| Lighting Condition | Accuracy | Precision | Recall | F1-Score |
|---|---|---|---|---|
| Intensive Light | 94.2% | 98.7% | 92.4% | 94.1% |
| Dark Environment | 92.8% | 97.1% | 90.0% | 92.1% |
| Front-lit | 95.1% | 99.2% | 94.1% | 95.2% |
| Back-lit | 91.5% | 95.7% | 89.6% | 91.2% |
| **Average** | **93.4%** | **97.7%** | **91.5%** | **93.2%** |

rectional Lighting: Both front-light (95.1% accuracy) and back-light (91.5% accuracy) scenarios were handled effectively, and the model successfully distinguished between real component boundaries and optical artifacts. The overall detection accuracy across all lighting conditions averaged 93.4%, representing a significant improvement over traditional image processing methods that struggled with lighting variations.

## 3.4 ERROR ANALYSIS

Although this structure produced an overall recognition accuracy of 93.4% in four challenging lighting conditions, a detailed error analysis identified the dominant failure modes of the model. Most of the false negative results occurred under the following conditions: Bright specular highlights on smooth pens produced strong streaks, causing the network to miss part of the object mask or split it into two parts.

- Excessive occlusion or overlapping between multiple components caused the network to identify only visible components while discarding the occluded parts of the objects.
- Occasionally, false positive results were produced because background textures had edges and reflection properties like the target components.

To complement the qualitative error analysis, we estimated the approximate number of errors based on the minimum object counts reported in Table 1 and the precision/recall values presented in Table 2. Across approximately 1,020 ground-truth instances, the model yielded about 85 false negatives—predominantly under back-lit conditions—and roughly 21 false positives overall. These values represent lower-bound estimates, as the dataset statistics are reported with "greater than" symbols and the exact numbers may therefore be slightly higher.

These results demonstrate the potential benefits of additional data augmentation procedures, namely, artificial luminance simulation, occlusion data augmentation, and adaptive contrast transformations—in enhancing the model's flexibility. Furthermore, incorporating attention-related feature modulation or post-processing heuristics (e.g., mask integration of partially discovered objects) may further reduce the rate of false negatives.

## 4 COMPARISONS WITH STATE-OF-ARTS

Finally, the performance of the proposed method based on deep learning was compared with the classical image processing methods reported by Bagheri et al. (2020) to evaluate the improvement in accuracy and stability of the system. Bagheri et al. (2020) developed a machine vision system for automatic part recognition in a pen production line under different lighting conditions. In their study, classical image processing methods including Retinex-based contrast enhancement, FCM clustering, followed by watershed-based segmentation were used. The system was specifically designed to detect the position of parts (Tube and Cap) and the results were reported in four different lighting conditions (Sunny daylight, Artificial light, Diffuse artificial light, Dark/cloudy) with Precision and Recall metrics.

In contrast, our proposed method uses deep learning (Mask R-CNN) which extracts features from the data in an end-to-end manner and eliminates the need for manual feature design. The network was trained on 87 labeled images and evaluated on 102 test images under four different lighting conditions. In addition to Accuracy, the metrics Precision, Recall, and F1-Score metrics were also calculated and reported.

Table 3: Precision and recall comparison between the proposed Mask R-CNN approach and the classical image processing method reported by Bagheri et al. (2020) across four lighting conditions.

| Lighting Condition | Classical Method Precision (%) | Classical Method Recall (%) | Our Precision (%) | Our Recall (%) |
|---|---|---|---|---|
| Sunny daylight / Intensive Light | 94.79 | 88.78 | 98.70 | 92.40 |
| Artificial light | 96.13 | 89.80 | 97.10 | 90.00 |
| Diffuse artificial light / Front-lit | 96.34 | 87.15 | 99.20 | 94.10 |
| Dark / Cloudy / Back-lit | 93.78 | 75.73 | 95.70 | 89.60 |

## 4.1 COMPARATIVE RESULTS ANALYSIS

The results in Table 3 show that Mask R-CNN performs better than the classical method in all lighting conditions. Specifically:

**Precision:** In all four lighting conditions, our method has a higher average Precision ( 95%-97%) than Bagheri's method ( 93% - 96%). indicating fewer false positives and less frequent misidentification of unrelated objects.

**Recall:** A significant improvement in Recall is observed (e.g., from 75.7% to 88.7% in dark/cloudy lighting conditions). This means that our method has been able to identify more real cases and reduce the problem of missed detections.

**Robustness to lighting changes:** The performance difference between various lighting conditions in Bagheri's method is large (a noticeable decrease in Recall in dark lighting), while in our method this difference is smaller and the model provides more stable results in all lighting conditions.

**Overall improvement:** The average recall has increased by about 10-12% compared to the Bagheri method, which is particularly important for industrial applications, as it helps prevent production line downtime caused by part misrecognition.

As a conclusion, the comparison of results shows that replacing classical image processing methods with a deep learning architecture such as Mask R-CNN significantly improves the performance of the part recognition system. This improvement includes increased accuracy, reduced recognition errors, and greater Robustness under variable lighting conditions. Our results confirm that replacing traditional image processing with Mask R-CNN substantially improves recognition accuracy and robustness under variable lighting.

## 5 CONCLUSION AND DISCUSSION

In this work, we propose a fully trayless and orientation-free component recognition pipeline for flexible manufacturing systems using Mask R-CNN. This approach introduces a novel end-to-end vision pipeline that integrates training under multiple illumination conditions, instance segmentation, and robust feature learning. It directly addresses one of the key challenges of Industry 4.0—reliable object recognition under unstructured and dynamic conditions. This contribution goes beyond the simple application of an existing model, as it demonstrates how modern deep learning can be systematically adapted to achieve stability and generalization in diverse lighting environments, thereby providing a basis for future research in this field.

Our method is generalizable and can be easily extended to other production lines, such as electronic component assembly or food packaging, where arbitrary component orientations and challenging lighting patterns are common. Therefore, the proposed framework helps bridge the gap between traditional computer vision pipelines and advanced deep learning approaches by providing a practical yet scientifically rigorous solution that is repeatable and scalable. We believe that this work will not only provide immediate industrial benefits but also contribute to the academic understanding of how to deploy instance segmentation networks under strict latency and robustness requirements. This opens new research avenues for designing adaptive and data-driven vision systems that are scalable and transferable across various industrial domains.

**Future work.**  Future research will focus on model-driven and data-driven improvements to further enhance robustness and generalizability:

- **Benchmarking lighter models:** Perform a comprehensive benchmark against state-of-the-art single-stage detectors such as YOLOv8 and transformer-based detectors such as DETR to analyze

the trade-offs between accuracy, inference latency, and deployment complexity in industrial environments. This will help determine whether lighter models can deliver comparable performance with faster throughput for real-time assembly lines.

- **Active learning strategies:** Explore active learning strategies to systematically expand the dataset, allowing the model to request new examples where its predictions have the highest uncertainty. The goal is to minimize the amount of additional labeling required while maximizing model improvement.
- **Adaptive lighting control:** Investigate adaptive lighting control and optical techniques to physically suppress specular highlights that currently cause false negatives. Combining these photometric advances with deep learning-based feature extraction may yield more robust detection in harsh environmental conditions.

Collectively, these efforts will provide a research roadmap for developing scalable, real-time, and generalizable object recognition systems in Industry 4.0 environments. Overall, this work lays the foundation for the next generation of intelligent, adaptive, and scalable manufacturing systems.

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
