# OpenReview forum: "ROBUST COMPONENT DETECTION FOR FLEXIBLE MANUFACTURING: A DEEP LEARNING APPROACH TO TRAY-FREE OBJECT RECOGNITION UNDER VARIABLE LIGHTING"
_ICLR.cc/2026/Conference — ICLR 2026 Conference Desk Rejected Submission_

### Official Review · Reviewer_F22s · 2025-10-17

**Soundness:** 2
**Presentation:** 1
**Contribution:** 1
**Rating:** 0
**Confidence:** 5

**Summary:**

This paper aims to address a real-world object detection problem in a pen production line. The evaluation shows promising results in the considered experimental setup. This work reads like an unfinished paper with very limited contributions. It is not ready for being published.

**Strengths:**

This paper targets a concrete, real-world object detection problem, yielding promising results within the considered evaluation pipeline.

**Weaknesses:**

1. The writing of the paper needs to be revised significantly. The paper reads like a barely finished report, not even a technical one.
2. The novelty and the contributions of the work are very limited.
3. This work could potentially contribute some insightful findings after re-designing the whole experiment and rewriting the whole paper. However, I couldn't find any useful insights from this work in its current form.

**Questions:**

None

---

### Official Review · Reviewer_R3bR · 2025-10-22

**Soundness:** 3
**Presentation:** 3
**Contribution:** 1
**Rating:** 2
**Confidence:** 4

**Summary:**

The paper presents an approach to detect pen components in an assembly line under various light conditions, achieving an accuracy of up to 95%.

**Strengths:**

- The paper tackles a practically relevant problem
- Evaluation of the approach under various lighting conditions
- The problem setting is described clearly

**Weaknesses:**

- I am surprised to see this paper at ICLR and I am wondering whether there wouldn't be a more suitable venue
- The approach seems very practical, but the scientific contribution does not become sufficiently clear
- (Strong) baselines are missing (e.g., other models apart from Mask R-CNN)
- The dataset seems rather small (Table 1) and its description seems inconsistent: Why are there 8 images of 87 objects? Were the column headings mixed up? In Section 3.1, it is mentioned that 22 images were used for training. How is this inconsistency to Table 1 explained?
- The focus on pen manufacturing seems to be rather limited and a wider area of applications would be desirable.
- Some typos/formatting issues (e.g., missing space after comma, wrong quotation marks, ...)

**Questions:**

- How many training images were used 8, 22 or 78?
- Can you include stronger baselines?
- What is your scientific contribution compared to previous related work?

---

### Official Review · Reviewer_Uh26 · 2025-10-28

**Soundness:** 1
**Presentation:** 3
**Contribution:** 1
**Rating:** 0
**Confidence:** 5

**Summary:**

This paper investigated to train and deploy a Mask R-CNN for detect pens in an industrial pen production line for robot to pick up and assembly. The model is trained with a few images and then tested on a separated test set. In comparison with SOTA, it is only compared with classical machine learning method.
The investigated technology is out-of-date. First, it does not show why the task is challenging for SOTA Vision Foundation Models (e.g., SAM, GroundingDINO). The listed challenging conditions like lighting and orientation are not challenging to recent approaches for OOD, Domain Adaption, and Image Augmentation. No related works presented in this paper. The presented technical part is just re-training an existing model, no novel technology is proposed to address clear challenges.

**Strengths:**

an experiment on industrial data.

**Weaknesses:**

no deep study, no novelty.

**Questions:**

An investigation with recent progresses could be performed first, such as VFM for zero-shot learning, general model adaption approaches like Image Augmentation, Domain Adaption, to show if the task is really challenging to SOTA approaches, and investigating novel and better approach to address the issues.

---

### Official Review · Reviewer_Fnm9 · 2025-10-29

**Soundness:** 2
**Presentation:** 3
**Contribution:** 1
**Rating:** 0
**Confidence:** 4

**Summary:**

The authors apply Mask RCNN to the task of industrial pen component recognition in variable environmental conditions. They deploy the model in a real-world industrial environment, achieving solid performance.

**Strengths:**

1. The paper is generally well written and organized.
2. The improvements over the classical method (Bagheri et al. 2020) are solid, particularly for dark conditions (Table 3).
3. The real-world implementation is commendable, and not something that you often see in ICLR submissions.

**Weaknesses:**

My biggest concern for this paper is that it doesn't seem to be in-scope for ICLR. There are no technical developments in deep learning, machine learning, etc, as the authors are just applying an existing method, Mask R-CNN, unchanged, to a new and very niche application domain. On that note, the evaluation is quite limited, on a single small dataset, with no comparison to other learning-based detection methods (old or modern).

Overall, because of the lack of any technical developments in deep learning, machine learning, or computer vision, and the very limited and niche scope of the evaluated application scenario, I don't see how this would would be of interest to the broader ICLR community.

**Minor Weaknesses**
1. The citations for R-CNN, Fast R-CNN, Faster R-CNN, YOLO and SSD in the introduction are not for the original corresponding method papers, but for applications of these methods to industrial detection. Please properly cite these methods from their original papers, and re-locate the applied paper citations elsewhere.
2. The citation for Profili et al. 2025 in paragraph 2 of the introduction has an erroneous period right before it.

**Questions:**

Is there any argument or new experimental evidence which the authors can provide, that would increase the interest of this work to the broader ICLR/machine learning/computer vision community? Unfortunately, I don't know if any revision or rebuttal to the work would make it appropriate for ICLR, as it is a very applied work, in a very niche domain, using an existing popular detection method with no technical novelty (and only comparing to a single classical method; no modern deep learning-based techniques).

---

### Official Review · Reviewer_oBFK · 2025-10-31

**Soundness:** 3
**Presentation:** 3
**Contribution:** 1
**Rating:** 0
**Confidence:** 5

**Summary:**

This paper designed a Mask R-CNN-based object detection method for pen production line. It also made a computer vision system which can help industrial robots to detect and pick up tray-free components. It can achieve 95% recognition accuracy under variable lighting environment.

This paper should be rejected because:
1) its engineering value is good, however, its academic contribution to this area is quite limited,  its Mask R-CNN-based method is not good enough; 2) the experimental comparison is quite limited.

**Strengths:**

1. The technical details are sound and can be reproduced.
2. The introduced vision system can work well in variable lighting conditions and bring significant engineering value.

**Weaknesses:**

1. The author should identify an influential academic point to write this paper. I do not think tray-free object detection and pick up algorithm is still a challenging problem in industrial environments. However, difficulty in generalization ability of object detection, or in detecting high light-reflection industrial parts, etc., are well-known challenging problems.
2. The novelty of Mask R-CNN-based model should be identified and improved.
3. There are too many object detection methods, such as YOLO series, Faster RCNN, large model-based, etc. these related studies should be reviewed completely.
4. The comparison is not enough.

**Questions:**

1. Failure cases can be visualized, and give deep analysis why they are failed. It is quite useful to find the major patterns in your pen production line. And it may give some insights to design data-suitable detection model.

---

### Note · Program_Chairs · 2026-01-17
**Submission Desk Rejected by Program Chairs**

The following references in this submission do not refer to real documents and/or have major errors in bibliographic information:

 A. Profili et al. Machine vision system for automatic defect detection of machined components using deep learning algorithms. International Journal of Advanced Manufacturing Technology, 125(7): 4023–4038, 2024.
A. Simeth et al. Flexible and robust detection for assembly automation with yolo in an hmlv production line under different lighting conditions. International Journal of Production Research, 62(18):5400–5418, 2024.